# Rates, causes and predictors of all-cause and avoidable mortality in 163 686 children and young people with and without intellectual disabilities: a record linkage national cohort study

Laura Anne Hughes-McCormack [ID] ,[1] Ewelina Rydzewska,[2] Sally-Ann Cooper [ID] ,[1] Michael Fleming,[3] Daniel Mackay,[3] Kirsty Dunn,[1] Laura Ward,[1] Filip Sosenko,[1] Fiona Barlow,[1] Jenny Miller,[4] Joseph D Symonds,[1] Bhautesh D Jani,[5] Maria Truesdale [ID] ,[1] Deborah Cairns,[1] Jill Pell,[3] Angela Henderson [ID] ,[1] Craig Melville[1]

For numbered affiliations see end of article.

**Correspondence to**
Laura Anne Hughes-McCormack;
Laura.Hughes-Mccormack@glasgow.ac.uk

## ABSTRACT

**Objectives** To investigate mortality rates and associated factors, and avoidable mortality in children/young people with intellectual disabilities.

**Design** Retrospective cohort; individual record-linked data between Scotland's 2011 Census and 9.5 years of National Records for Scotland death certification data.

**Setting** General community.

**Participants** Children and young people with intellectual disabilities living in Scotland aged 5–24 years, and an age-matched comparison group.

**Main outcome measures** Deaths up to 2020: age of death, age-standardised mortality ratios (age-SMRs); causes of death including cause-specific age-SMRs/sex-SMRs; and avoidable deaths.

**Results** Death occurred in 260/7247 (3.6%) children/young people with intellectual disabilities (crude mortality rate=388/100 000 person-years) and 528/156 439 (0.3%) children/young people without intellectual disabilities (crude mortality rate=36/100 000 person-years). SMRs for children/young people with versus those without intellectual disabilities were 10.7 for all causes (95% CI 9.47 to 12.1), 5.17 for avoidable death (95% CI 4.19 to 6.37), 2.3 for preventable death (95% CI 1.6 to 3.2) and 16.1 for treatable death (95% CI 12.5 to 20.8). SMRs were highest for children (27.4, 95% CI 20.6 to 36.3) aged 5–9 years, and lowest for young people (6.6, 95% CI 5.1 to 8.6) aged 20–24 years. SMRs were higher in more affluent neighbourhoods. Crude mortality incidences were higher for the children/young people with intellectual disabilities for most International Statistical Classification of Diseases and Related Health Problems, 10th Revision chapters. The most common underlying avoidable causes of mortality for children/young people with intellectual disabilities were epilepsy, aspiration/reflux/choking and respiratory infection, and for children/young people without intellectual disabilities were suicide, accidental drug-related deaths and car accidents.

**Conclusion** Children with intellectual disabilities had significantly higher rates of all-cause, avoidable, treatable

## STRENGTHS AND LIMITATIONS OF THIS STUDY

⇒ Novel use of census records and record linkage to death records to study mortality in a total population cohort of children and young people with intellectual disabilities.
⇒ Due to the use of a whole country population, these results are well powered and generalisable.
⇒ Despite comprising a whole country population, our study was not large enough to delineate cause-specific mortality ratios by sex.
⇒ This study was limited by lack of demographic and clinical diagnostic information, including the severity or cause of intellectual disabilities.
⇒ Reliance on death certificate data is limited by inconsistencies in reporting of cause of death.

and preventable mortality than their peers. The largest differences were for treatable mortality, particularly at ages 5–9 years. Interventions to improve healthcare to reduce treatable mortality should be a priority for children/young people with intellectual disabilities. Examples include improved epilepsy management and risk assessments, and coordinated multidisciplinary actions to reduce aspiration/reflux/choking and respiratory infection. This is necessary across all neighbourhoods.

## INTRODUCTION

Children and young people with intellectual disabilities have a significantly higher prevalence of physical and mental ill health compared with the general population.[1–3] The life expectancy of people with intellectual disabilities has been reported to be shorter, on average 20 years younger than in the general population, although this may be substantially lower in some

countries such as America,[4] including deaths considered potentially avoidable.[5][6]

Few studies have reported on mortality specifically in children and young people with intellectual disabilities.[7–22] A systematic review highlighted that many studies lacked baseline data on sex and age, and not all report age-specific death rates,[5] while very few report on cause of death, or on avoidable deaths. Two of the studies focused only on young people, aged 18+[22] and 20+.[9] A few large-scale data linkage studies have investigated mortality in children and young people with and without intellectual disabilities.[7][12][19] One study used Scotland's Pupil Census records linked to National Records of Scotland (NRS) Statutory Register of Deaths (from 2008 to 2013)[7] and found standardised mortality ratios (SMR) were substantially higher in children and young people with intellectual disabilities compared with those without (SMR=11.6, 95% CI 9.6 to 14.0). SMR was higher for children aged 5–14 years (SMR=21.6, 95% CI 16.6 to 28.2) than young people aged ≥15 years (SMR=7.7, 95% CI 5.9 to 10.2), and for females. However, this study used the broad definition of intellectual disabilities employed in Scottish schools, requiring a sensitivity analysis around which children to include in the analyses. A similar pattern, though to a lesser extent, was found in a study using data from the Western Australian Intellectual Disability Exploring Answers Database linked to the Western Australian Mortality Database (from 1983 to 2010).[14] Children and young people with intellectual disabilities aged 1–25 years had a higher risk of death (adjusted HRs (aHR)=6.1, 95% CI 5.3 to 7.0), compared with children without intellectual disabilities. aHR for mortality was higher for children aged 6–10 years (aHR=12.6, 95% CI 9.0 to 17.7) than for those aged 11–25 years (aHR=4.9, 95% CI 3.9 to 6.1).[11] A study from Ireland[19] reported that mortality was almost seven times higher among children and young people aged 0–19 years in the intellectual disabilities population than the general population (SMR=6.68, 95% CI 5.91 to 7.52). However, this study used a restrictive definition of intellectual disabilities since identification was carried out using a database of children and young people known to intellectual disabilities services (the National Intellectual Disabilities Database). These children and young people were likely to have higher care needs and comorbidities associated with premature death than a broader group of people with intellectual disabilities. The control general population group was obtained from a different database (Irish Central Statistics Office), and this did not include a marker for intellectual disabilities. Despite their limitations, each of these studies reported SMRs to be substantially higher for children and young people with intellectual disabilities, indicating pervasive health inequalities may be contributing to avoidable deaths in childhood.

Few studies report data on causes of death in children and young people with intellectual disabilities.[7][12][14] The findings of previous research are inconsistent due to varying methodologies,[5] and most cause-specific mortality findings have been grouped across all childhood ages due to small sample sizes.[7][14] Other limitations are failure to report cause-specific SMRs by International Statistical Classification of Diseases and Related Health Problems, 10th Revision (ICD-10) chapters.[12] It is clear that robust research is needed to further elucidate causes of death in the population of children and young people with intellectual disabilities, and to identify possible interventions to address this health inequality.

It is not clear whether children and young people with intellectual disabilities experience avoidable deaths more commonly than other children and young people. The Office for National Statistics (ONS) defines avoidable deaths as either 'treatable' (previously known as 'amenable') with timely and effective healthcare, 'preventable' through public health action or both.[23][24] Only three previous studies have reported on avoidable mortality among children and young people with intellectual disabilities (one of which only focused on young people aged 18+[22] and one of which did not present numerical data)[7][8][22] and only two have reported on deaths from treatable mortality.[7][22] These studies all found higher rates of deaths from avoidable or treatable mortality in the intellectual disabilities population. A data linkage study using the Scotland's Pupil Census found avoidable mortality was approximately 3.6 times higher for children and young people with intellectual disabilities compared with peers, although this figure was based on low numbers and was therefore classed as 'unreliable' by the authors.[7] There is a need to quantify the extent and patterns of avoidable mortality in children and young people with intellectual disabilities compared with general population peers using large and valid data sets.

The aim of this study is to investigate deaths in children and young people with and without intellectual disabilities from 2011 to 2020 using data from Scotland's 2011 Census linked to the NRS Statutory Register of Deaths. Specifically, we investigated (a) the age-SMR and sex-SMR for children and young people with intellectual disabilities, (b) the common causes of death for children and young people with intellectual disabilities, and any differences compared with peers, (c) the proportion of deaths considered avoidable (including deaths from treatable and preventable mortality) for children and young people with intellectual disabilities, and any differences compared with peers, and (d) whether factors (such as socioeconomic and demographic factors) are associated with deaths in the population with intellectual disabilities.

## METHODS
### Patient and public involvement

This study was undertaken in the Scottish Learning Disabilities Observatory due to growing concern among people with intellectual disabilities and their families around mortality. The steering group included people with intellectual disabilities and partners from third

sector organisations. This project was carried out in collaboration with an organisation in Scotland that works solely with people with profound and multiple learning disabilities and their families for a better life (Promoting a More Inclusive Society, PAMIS). Results from this study will be disseminated to people with intellectual disabilities and their families in an easy-read version via the Scottish Learning Disabilities Observatory/PAMIS websites and via a range of other communication methods (such as blogs/newsletters).

## Study sample, setting and process

We used data from Scotland's 2011 Census to create a cohort of children and young people with intellectual disabilities, aged 5–24 years at the census date, and comparison group matched for age (identified from a larger population consisting of a 15% random and unmatched sample of the Scottish population also identified from the 2011 Census, with neither intellectual disabilities nor autism). Full details of the methodology and other background information on Scotland's 2011 Census are available at http://www.scotlandscensus.gov.uk/supporting-information. We used the NRS Indexing Service to link the census data to the NRS Indexing Spine, which includes each person's Community Health Index (CHI). The CHI is a unique National Health Service identifier given to everyone in Scotland. Indexing enabled linkage to a range of health databases, including the NRS Statutory Register of Deaths database, to ascertain all deaths up to 15 August 2020. Access to the anonymised linked data was made available to approved members of our team via Scotland's National Safe Haven.

## Data sources and definitions
### Identification of children and young people with intellectual disabilities

Scotland's 2011 Census provides statistical information on the number and characteristics of Scotland's population and households at the census day, 27 March 2011. It includes people living in communal establishments (such as care homes and student halls of residence) as well as people living in private households. In 2011, the census in Scotland was estimated to have achieved a 94% response rate (http://www.scotlandscensus.gov.uk/supporting-information). Scotland's Census is one of few country censuses that identify people with intellectual disabilities and distinguish these individuals from people with specific learning difficulties such as dyslexia; indeed, it may be unique in this regard. Full details of the methodology and other background information on Scotland's 2011 Census are freely available online.[25] The Census requires the form to be completed by the head of household or joint head of household on behalf of all occupants in private households, and the manager is responsible on behalf of all occupants in communal dwellings. It is a legal requirement to complete the census, and the census form clearly states this. A head of household

not completing the census or supplying false information can be fined £1000. The census team follow-up non-responders and provide help to respond when required, hence the high 94% completion rate. Self-reporting/proxy reporting was used to identify children and young people with intellectual disabilities from question 20: 'Do you have any of the following conditions which have lasted, or are expected to last, at least 12 months? Tick all that apply'. Respondents were given a choice of 10 response options: (1) deafness or partial hearing loss, (2) blindness or partial sight loss, (3) learning disability (eg, Down syndrome), (4) learning difficulty (eg, dyslexia), (5) developmental disorder (eg, autistic spectrum disorder or Asperger's syndrome), (6) physical disability, (7) mental health condition, (8) long-term illness, disease or condition, (9) other condition, and (10) no condition. Importantly, the question distinguished between intellectual disabilities (for which the term 'learning disability' is used in the UK, as in option 3), learning difficulty (which in the UK is synonymous with the international term 'specific learning disability' such as dyslexia) and autism. It is important to note that as multiple response options could be selected from the conditions list, the population with intellectual disabilities does overlap with the population with autism (as well as other conditions) and will include various different types of intellectual disabilities (eg, individuals with Down syndrome or cerebral palsy (in cases where cerebral palsy co-occurs with intellectual disabilities)). This is however common in research of this nature as these two conditions often co-occur and there is also a range of causes of intellectual disabilities. The proportion of the population with intellectual disabilities who also reported co-occurring autism is reported in the Results section. In the comparison population, individuals who reported autism (without also reporting co-occurring intellectual disabilities) were removed from the analysis as this population also experiences a different health profile and substantial health inequalities compared with the general population, which would likely influence the mortality outcomes in the comparison population. It is also important to note that intellectual disabilities may or may not co-occur with cerebral palsy. It is likely that only individuals with cerebral palsy and intellectual disabilities would select the option for 'intellectual disabilities' in the census, and most likely in combination with 'physical disability'. Those with cerebral palsy with no intellectual disability may be more likely to select 'physical disability' (or possibly 'other condition'). However, this level of detail about the reporting was not available.

## Age

Grouped into four age categories of (1) 5–9 years, (2) 10–14 years, (3) 15–19 years, and (4) 20–24 years.

## Scottish Index of Multiple Deprivation

Scottish Index of Multiple Deprivation (SIMD) is derived from individual postcode of residence and calculated at data zone level. SIMD is a composite of seven indices and over 30 indicators to indicate the extent of neighbourhood deprivation. SIMD was divided into quintiles according to the general population where SIMD 1 represents the most deprived neighbourhoods and SIMD 5 represents the most affluent neighbourhoods.[26–28]

## Deaths

In Scotland, it is a legal requirement that all deaths are notified by the responsible clinician by completing a death certificate. These are registered at NRS. Using the ICD-10 codes[29] according to death certificates registered at NRS, we identified deaths for people with intellectual disabilities and the general population comparison group from dates of death recorded on the death certificates. For cause of death analyses, the underlying cause of death is defined internationally[29] as the disease or injury which initiated the chain of morbid events leading directly to death, or the accident/act which produced the fatal injury. We also used a broader definition to analyse all-contributing causes, that included all deaths, with any mention on the death certificate related to the cause; combining both the underlying causes with secondary or contributing factors. While the same ICD-10 codes are used, it is important to note that one death may have several other additional causes as contributing factors, all of which are counted in figures reporting 'all-contributing causes'. We defined treatable and preventable deaths from avoidable mortality outcomes outlined in the guidance of the ONS,[23 24] and defined diagnostic ICD-10 codes in death certificates.[29] Some causes of death are both treatable with medical treatment and preventable through effective healthcare, and these are not mutually exclusive categories. The analyses were restricted to deaths recorded between the census date 27 March 2011 and 15 August 2020.

## All follow-up/censoring

Children were followed up from the census date 27 March 2011, and all models were censored on death or 15 August 2020 (whichever came first).

## Missing data

Data linkage was conducted by NRS, and all data provided to us for this study included complete cases only. We included all the cases provided from NRS in the analysis. The only exception to this was if there was a date of death that came prior to the date of the census—these cases were excluded. Errors in cause of death records such as omission and use of abbreviations were listed as an unknown cause. All deaths where the underlying cause was ill defined or defined by ICD-10 WHO guidelines[30] as codes in chapter 18 were also reclassified as 'unknown'.

## Analyses

### Intellectual disabilities

Age, sex and SIMD were taken from the time of the census for the children and young people. Explorative statistical analyses including t-tests and $\chi^2$ tests were used to investigate characteristics of children and young people with intellectual disabilities compared with peers in the general population. Differences in age at death were explored (using the median and IQR).

### Deaths

Crude mortality rates (CMR) per 100 000 were calculated using the censor date/date of death. For indirect standardisation, observed deaths were assumed to be independent and vary with the Poisson distribution. The mortality rates were indirectly standardised for both males and females using the expected age-specific mortality rates per 1-year age group using Stata's 'strate' command, to calculate age-SMRs for pupils with versus without intellectual disabilities. The 95% CIs were calculated based on the quadratic approximation of the log likelihood. Expected rates were calculated using fixed age and sex-specific rates from the large control population. The SMRs were subsequently calculated stratified by age (into ages 5–9, 10–14, 15–19 and 20–24 years), sex and SIMD. The SMRs were also calculated for all deaths. For all-cause mortality, Kaplan-Meier survival curves were plotted for the overall time for both groups and the proportional hazards assumption was tested. For the underlying causes of death, the total number of deaths in each ICD-10 chapter was collated, and this was then repeated for specific causes listed within chapters. Next, the breakdown of all-contributing causes was analysed by collating the number of deaths in each ICD-10 chapter. For cause-specific SMRs, indirect age standardisation was performed using 5-year age bands to age standardise rates. Robust SEs were used. The rates and age-SMRs for avoidable, treatable and preventable mortality were calculated using robust errors. Cox proportional hazards models were fitted to the data to calculate risks of mortality (all, avoidable, treatable, preventable) unadjusted and adjusted for age, sex and SIMD. For categories with fewer than 10 deaths, no calculation was attempted due to lack of reliability. Furthermore, in keeping with the ONS mortality methodology,[23] all mortality rates between 10 and 20 deaths were labelled as unreliable. One researcher (LAH-M) carried out the main analyses and a second researcher (ER) verified these for accuracy. All analyses were conducted in Stata V.14.

## RESULTS

Of the people with intellectual disabilities recorded in Scotland's 2011 Census, 22 538 (92.9%) were successfully linked with their health records. Regarding the control population who had neither intellectual disabilities nor

**Table 1** Demographic information for children and young people aged 5–24 years at baseline with and without intellectual disabilities

| Demographic information* | Intellectual disabilities | | Controls | | P value† |
|---|---|---|---|---|---|
| Total, n (person-years) | 7247 | 66 871.403 | 156 439 | 1 466 631.0 | – |
| Male sex, n (%) | 4460 | 61.5 | 77 979 | 49.8 | <0.001 |
| Age, n (%) | | | | | |
| 5–9 | 1435 | 19.8 | 34 607 | 22.1 | <0.001 |
| 10–14 | 1879 | 25.9 | 37 514 | 24.0 | |
| 15–19 | 2063 | 28.5 | 40 990 | 26.2 | |
| 20–24 | 1870 | 25.8 | 433 328 | 27.7 | |
| SIMD quintile, n (%) at time of census | | | | | |
| 1 (most deprived) | 1781 | 24.6 | 30 868 | 19.7 | <0.001 |
| 2 | 1522 | 21.0 | 29 765 | 19.0 | |
| 3 | 1417 | 19.6 | 30 742 | 19.7 | |
| 4 | 1343 | 18.5 | 31 387 | 20.1 | |
| 5 (least deprived) | 1184 | 16.3 | 33 677 | 21.5 | |
| Deaths, n, crude rate per 100 000 (CI)‡ | 260 | 388 (344–439) | 528 | 36 (33–39) | – |

*Data taken from time of census.
†$X^2$ test for intellectual disabilities compared with control group. (For SIMD, $X^2$ test (Pearson $\chi^2$ test for independence) was performed across all categories, overall p value.)
‡Three individuals had a record of death which occurred before the date of the census so were removed.
SIMD, Scottish Index of Multiple Deprivation.

autism, of the 15% randomly selected, 700 437 (95.1%) were successfully linked to their health records. The data sets included 7247 people with intellectual disabilities aged 5–24 years and 156 439 general population aged 5–24 years. Three individuals were excluded from the study cohort due to their date of death recorded as prior to census.

### Demographic information
Table 1 presents detailed demographic information on the population of children and young people with and without intellectual disabilities. The population of children and young people with intellectual disabilities had a higher proportion of males (as expected) than their peers (4460/7247; 61.5% vs 77 979; 49.8%, p<0.001). Up to 3029 (41.7%) of the population with intellectual disabilities had co-occurring autism (n=2037; 67.3% of whom were male). Children and young people with intellectual disabilities were more likely to be living in more deprived neighbourhoods (p<0.001) and were younger (p<0.001) than children and young people without intellectual disabilities.

### All-cause deaths
#### Mortality incidence
The study period (March 2011 to August 2020) resulted in the equivalent of 1 533 502 person-years of follow-up (this included 66 871 person-years contributed for the intellectual disabilities population and 1 466 631 person-years for the non-intellectual disabilities population). The median age at death for children and young people

with intellectual disabilities was younger at 19.5 years (SD=6.0; IQR=16–24) compared with 23.0 years (SD=5.0; IQR=19–27) for children and young people without intellectual disabilities.

Of the 7247 children and young people with intellectual disabilities, 260 (3.6%) had died during the 9.5 years of follow-up. Of the 156 439 children and young people without intellectual disabilities, 528 (0.3%) had died during the same follow-up period. Crude mortality incidence for the intellectual disabilities cohort during the period was 388 (344–439) per 100 000 person-years of follow-up, and 36 (33–39) per 100 000 for those without intellectual disabilities. Proportional hazards assumption was met (visually assessed). Kaplan-Meier survival curves for the overall time period were run.

### Standardised mortality ratios
Compared with the children and young people without intellectual disabilities, for all deaths, the SMR was 10.7 (9.5–12.1), 7.8 (6.6–9.1) for males and 18.1 (15.0–21.9) for females. The SMR was highest in the youngest age group (5–9 years) at 27.4 (20.6–36.3) and decreased with increasing age groups (10–14 years: 15.8 (12.7–19.8); 15–19 years: 8.6 (6.9–10.7); 20–24 years: 6.6 (5.1–8.6)). The SMR was highest for the most affluent SIMD level (26.3 (20.2–35.6)) and decreased with deprivation level (most deprived: 6.0 (4.6–7.9)). The SMRs are presented in table 2; specific numbers of deaths are not reported in this table due to some small numbers to prevent statistical

Table 2  Standardised mortality ratios (SMRs) for children and young people aged 5–24 years with intellectual disabilities compared with those without intellectual disabilities by age group, sex and deprivation (SIMD)

| Demographic variables | SMRs (all deaths) | 95% CI | Avoidable SMRs | 95% CI | Treatable SMRs | 95% CI | Preventable SMRs | 95% CI |
|---|---|---|---|---|---|---|---|---|
| Overall (age-SMR) | 10.7 | 9.5 to 12.1 | 5.2 | 4.2 to 6.4 | 16.1 | 12.5 to 20.8 | 2.3 | 1.6 to 3.2 |
| Age | | | | | | | | |
| 5–9 | 27.4 | 20.6 to 36.3 | 10.9 | 6.2 to 19.2 | 29.4 u | 13.2 to 65.5 | 6.7 u | 3.0 to 14.9 |
| 10–14 | 15.8 | 12.7 to 19.8 | 8.1 | 5.5 to 11.8 | 40.6 | 26.2 to 62.9 | 2.9 u | 1.5 to 5.7 |
| 15–19 | 8.6 | 6.9 to 10.7 | 3.7 | 2.5 to 5.5 | 10.8 u | 6.5 to 17.9 | 1.7 u | 0.9 to 3.2 |
| 20–24 | 6.6 | 5.1 to 8.6 | 4.1 | 2.8 to 6.1 | 11.4 u | 7.2 to 18.1 | 1.7 u | 0.9 to 3.2 |
| Sex | | | | | | | | |
| Male | 7.8 | 6.6 to 9.1 | 3.8 | 2.9 to 4.9 | 16.8 | 12.1 to 23.3 | 1.7 | 1.1 to 2.5 |
| Female | 18.1 | 15.0 to 21.9 | 8.3 | 5.9 to 11.6 | 15.7 | 10.4 to 23.6 | 3.7 u | 2.1 to 6.6 |
| SIMD | | | | | | | | |
| 1 (most deprived) | 6.0 | 4.6 to 7.9 | 3.0 | 1.9 to 4.7 | 8.4 u | 4.8 to 14.8 | 1.8 u | 0.9 to 3.3 |
| 2 | 7.8 | 5.9 to 10.3 | 6.4 | 4.4 to 9.2 | 16.0 u | 9.9 to 25.8 | 3.6 u | 2.1 to 5.9 |
| 3 | 10.7 | 8.2 to 14.0 | 3.8 | 2.3 to 6.4 | 12.8 u | 7.1 to 23.1 | – | – |
| 4 | 15.7 | 12.1 to 20.4 | 7.7 | 4.8 to 12.2 | 40.9 u | 23.2 to 71.9 | – | – |
| 5 (least deprived) | 26.8 u | 20.2 to 35.6 u | – | – | – | – | – | – |

SMR reported for ≥10 deaths; U rates based on 10–20 deaths labelled 'u' for unreliable.
SIMD, Scottish Index of Multiple Deprivation.

disclosure concerns. The Cox proportional hazards unadjusted (and adjusted (adj) for age, sex, SIMD) for risk of all-cause death were as follows: HR 10.7 (9.2–12.4) (adj HR 9.8 (8.5–11.4)). The unadjusted rate was the same as the SMR for all-cause risk of deaths although adjustment reduced the HRs slightly.

## Cause of death
### Mortality incidence
In the population with intellectual disabilities, the three most common underlying causes of death according to ICD-10 chapters were: diseases of the nervous system (n=87, CMR=130 (105–160)); congenital malformations, deformations or chromosomal abnormalities (n=53, CMR=79.2 (60.5–103)); and diseases of the respiratory system (n=20, CMR=29.9[U] (19.2–46.3)). In the control group, the three most common underlying causes of death were: external causes (n=278, CMR=18.9 (16.8–21.3)); symptoms, signs and abnormal clinical and laboratory findings (n=64, CMR=4.36 (3.41–5.57)); and neoplasms (n=59, CMR=4.02 (3.11–5.19)).

Looking at all-contributing factors, in the group with intellectual disabilities, diseases of the nervous and respiratory systems, and congenital malformations, deformations or chromosomal abnormalities were most commonly recorded with 213, 187 and 112 records, respectively. In the control group, external causes, injury, poisoning, certain other consequences of external causes as well as diseases of the circulatory system were most commonly recorded with 539, 299 and 118 records, respectively. Tables 3 and 4 present detailed information on the underlying and all-contributing causes of death in the population of children and young people with and without intellectual disabilities.

Among children and young people with intellectual disabilities the biggest causes of death (based on specific ICD-10 codes) were: cerebral palsy unspecified (19.2%); epilepsy unspecified (3.8%); and ill-defined and unknown cause (3.1%). For all-contributing causes the pattern was the same for the first two causes, with cerebral palsy and epilepsy being highest. The next highest causes were respiratory related, including pneumonitis due to food or vomit inhalation; respiratory failure unspecified; and pneumonia unspecified organism. Among control children and young people, the biggest causes of death were: self-harm by strangulation or suffocation (17.0%); ill-defined and unknown cause (11.2%); and narcotics and psychodysleptics (8.5%). For all-contributing causes the biggest causes were: self-harm by strangulation or suffocation; asphyxiation; and unknown cause.

### Standardised mortality ratios
Across all ICD-10 chapters, the SMRs showed no rates that were lower for the children and young people with intellectual disabilities compared with controls. Where they could be calculated, SMRs were high for most ICD-10 chapter groups of underlying causes, particularly so for diseases of the nervous system (SMR=73.2 (59.3–90.3)), respiratory system (SMR=40.8 (26.3–63.2)) and digestive system (SMR=36.9 (23.6–57.9)). For all-contributing causes, SMRs were highest overall for congenital malformations, deformations and chromosomal abnormalities (SMR=222 (181–273)), diseases of the nervous system

**Table 3** Underlying causes of death, all-contributing factors in death and cause-specific crude mortality rates per 100000 person-years by ICD-10 chapters for children and young people aged 5–24 years with and without intellectual disabilities

| ICD-10 chapter | Underlying cause of death | | | | | | | All-contributing factors in death | | | | | | |
| --- | --- | --- | --- | --- | --- | --- | --- | --- | --- | --- | --- | --- | --- | --- |
| | Intellectual disabilities | | | Controls | | | | Intellectual disabilities | | | Controls | | | |
| | n* | CMR | 95% CI | n* | CMR | 95% CI | SMR (95% CI) | n* | CMR | 95% CI | n* | CMR | 95% CI | SMR (95% CI) |
| Chapter 1. Certain infectious and parasite diseases (A00–B99) | 10 | 14.9 u | 8.0 to 27.7 | <5 | – | – | – | 24 | 34.3 | 22.8 to 51.7 | 13 | 0.8 u | 0.5 to 1.4 | 42.3 (28.1 to 63.7) |
| Chapter 2. Neoplasms (C00–D49) | 5 | – | – | 59 | 4.0 | 3.1 to 5.2 | – | 7 | – | – | 82 | 4.6 | 3.6 to 5.8 | – |
| Chapter 3. Diseases of the blood, blood-forming organs and immune mechanism (D50–D89) | <5 | – | – | – | – | – | – | 9 | – | – | 6 | – | – | – |
| Chapter 4. Endocrine, nutritional and metabolic diseases (E00–E89) | 14 | 20.9 u | 12.3 to 35.3 | 13 | 0.9 u | 0.5 to 1.5 | 23.4 (13.8 to 39.4) | 29 | 40.3 | 27.6 to 58.8 | 28 | 1.6 | 1.1 to 2.4 | 24.6 (16.9 to 35.9) |
| Chapter 5. Mental and behavioural disorders (F01–F99) | <5 | – | – | 8 | – | – | – | 25 | 35.8 | 24.0 to 53.5 | 69 | 4.70 | 3.7 to 6.0 | 7.5 (5.1 to 11.2) |
| Chapter 6. Diseases of the nervous system (G00–G99) | 87 | 130 | 105 to 160 | 26 | 1.77 | 1.2 to 2.6 | 73.2 (59.3 to 90.3) | 213 | 233 | 199 to 272 | 39 | 2.52 | 1.8 to 3.5 | 92.9 (79.4 to 108) |
| Chapter 7. Diseases of the eye and adnexa (H00–H59) | – | – | – | – | – | – | – | <5 | – | – | – | – | – | – |
| Chapter 8. Diseases of the ear and mastoid process (H60–H95) | <5 | – | – | – | – | – | – | <5 | – | – | – | – | – | – |
| Chapter 9. Diseases of the circulatory system (I00–I99) | 9 | – | – | 41 | 2.79 | 2.1 to 3.8 | – | 43 | 55.3 | 40.0 to 76.3 | 118 | 4.8 | 3.8 to 6.1 | 11.3 (8.2 to 15.6) |
| Chapter 10. Diseases of the respiratory system (J00–J99) | 20 | 29.9 u | 19.2 to 46.3 | 11 | 0.8 u | 0.4 to 1.4 | 40.8 (26.3 to 63.2) | 187 | 213 | 181 to 251 | 48 | 2.9 | 2.2 to 3.9 | 73.1 (62.0 to 86.1) |
| Chapter 11. Diseases of the digestive system (K00–K95) | 19 | 28.4 u | 18.1 to 44.5 | 12 | 0.8 u | 0.5 to 1.4 | 36.9 (23.6 to 57.9) | 72 | 53.8 | 38.8 to 74.6 | 29 | 1.4 | 0.9 to 2.1 | 42.1 (30.4 to 58.4) |
| Chapter 12. Diseases of the skin and subcutaneous tissue (L00–L99) | <5 | – | – | – | – | – | – | <5 | – | – | <5 | – | – | – |
| Chapter 13. Diseases of the musculoskeletal system and connective tissue (M00–M99) | <5 | – | – | <5 | – | – | – | 20 | 28.4 u | 18.1 to 44.5 | 6 | – | – | – |
| Chapter 14. Diseases of the genitourinary system (N00–N99) | 5 | – | – | <5 | – | – | – | 13 | 20.9 u | 12.3 to 35.3 | 9 | – | – | – |
| Chapter 15. Pregnancy, childbirth and puerperium (O00–O9A) | – | – | – | <5 | – | – | – | – | – | – | <5 | – | – | – |
| Chapter 16. Certain conditions originating in the perinatal period (P00–P96) | <5 | – | – | – | – | – | – | <5 | – | – | – | – | – | – |
| Chapter 17. Congenital malformations, deformations and chromosomal abnormalities (Q00–Q99) | 53 | 79.2 | 60.5 to 103 | 6 | – | – | – | 112 | 137 | 112 to 168 | 10 | 0.6 u | 0.3 to 1.2 | 222 (181 to 273) |
| Chapter. 18. Symptoms, signs and abnormal clinical and laboratory findings (R00–R99) | 9 | – | – | 64 | 4.4 | 3.4 to 5.6 | – | 64 | 82.2 | 63.1 to 107 | 90 | 6.0 | 4.9 to 7.4 | 13.5 (10.4 to 17.6) |

Continued

**Table 3** Continued

| ICD-10 chapter | Underlying cause of death | | | | | | | All-contributing factors in death | | | | | | |
| --- | --- | --- | --- | --- | --- | --- | --- | --- | --- | --- | --- | --- | --- | --- |
| | Intellectual disabilities | | | Controls | | | SMR (95% CI) | Intellectual disabilities | | | Controls | | | SMR (95% CI) |
| | n* | CMR | 95% CI | n* | CMR | 95% CI | | n* | CMR | 95% CI | n* | CMR | 95% CI | |
| Chapter 19. Injury, poisoning and certain other consequences of external causes (S00–T88) | – | – | – | – | – | – | – | 39 | 34.3 | 22.8 to 51.7 | 539 | 19.6 | 17.4 to 22.0 | 1.7 (1.1 to 2.6) |
| Chapter 20. External causes of morbidity and mortality (V00–Y99) | 16 | 23.9 u | 14.6 to 39.0 | 278 | 18.9 | 16.8 to 21.3 | 1.2 (0.8 to 2.0) | 38 | 52.3 | 37.5 to 72.8 | 299 | 20.0 | 17.8 to 22.4 | 2.6 (1.8 to 3.6) |
| Total number of deaths | 260 | | | 528 | | | | 260 | | | 528 | | | |

CMR/SMR reported for ≥10 deaths.
U rates based on 10–20 deaths labelled 'u' for unreliable.
*n<5 repressed due to statistical disclosure.
CMR, crude mortality rate; ICD-10, International Statistical Classification of Diseases and Related Health Problems, 10th Revision; SMR, standardised mortality ratio.

(SMR=92.9 (79.4–108)), diseases of the respiratory system (SMR=73.1 (62.0–86.1)) and digestive (SMR=42.1 (30.4–58.4)).

## Avoidable (treatable and preventable) deaths
### Mortality incidence
Of all deaths (n=260) among children and young people with intellectual disabilities, 88 (33.8%) were considered avoidable, 59 (22.7%) were treatable and 34 (13.1%) were preventable. Of all deaths (n=528) among children and young people without intellectual disabilities, 369 (69.9%) were considered avoidable, 80 (15.2%) were treatable and 326 (61.7%) were preventable. Despite the higher proportion of preventable deaths out of all deaths in the controls, the incidence rate for preventative deaths (as well as treatable deaths and overall avoidable deaths) remained significantly higher in the intellectual disabilities group. The specific incidence rates were as follows:

► Avoidable mortality incidence for the intellectual disabilities cohort during the period was 131 (106–162) per 100 000 person-years of follow-up, and 25 (22–27) per 100 000 for those without intellectual disabilities.
► Treatable mortality incidence for the intellectual disabilities cohort during the period was 88 (68–113) per 100 000 person-years of follow-up, and 5 (4–6) per 100 000 for those without intellectual disabilities.
► Preventable mortality incidence for the intellectual disabilities cohort during the period was 50 (36–71) per 100 000 person-years of follow-up, with 22 (19–24) per 100 000 for those without intellectual disabilities.

### Standardised mortality ratios
Compared with the children and young people without intellectual disabilities, the SMR for avoidable deaths overall was 5.2 (4.2–6.4). Treatable SMRs were much higher (16.1 (12.5–20.8)) than preventable SMRs (2.3 (1.6–3.2)), which were also high. Some avoidable (treatable and preventable) SMRs by age group, sex and SIMD were considered unreliable due to small numbers (reported in table 2) but the trends for this show that SMRs were higher in the youngest age groups, and highest overall for treatable deaths. Of note, treatable deaths were substantially higher for children and young people with intellectual disabilities at ages 10–14 years (40.6 (26.2–62.9)) compared with avoidable, treatable or preventable SMRs across other age groups. Avoidable and preventable SMRs were higher for females (avoidable: 8.3 (5.9–11.6), preventable: 3.7[U] (2.1–6.6)) than males (avoidable: 3.8 (2.9–4.9), preventable: 1.7 (1.1–2.5)). However, the opposite was found for treatable deaths, where SMRs were higher for males (16.8 (12.1–23.3)) compared with females (15.7 (10.4–23.6)). There was a gradual increase in the SMRs with decreasing deprivation (SIMD) levels (although it should be noted that with decreasing deprivation levels, the numbers of deaths gradually got smaller—with many of these being considered 'unreliable'). Details are shown in table 2. The Cox proportional hazards unadjusted (and adjusted (adj) for

Table 4  Top causes of death in children and young people with and without intellectual disabilities (individual ICD-10 codes)

| Order | Underlying causes of deaths | | All-contributing factors in deaths | |
|---|---|---|---|---|
| | Intellectual disabilities (n) | Without intellectual disabilities (n) | Intellectual disabilities (n) | Without intellectual disabilities (n) |
| 1 | Cerebral palsy unspecified (50) | Intentional self-harm by strangulation and suffocation (90) | Cerebral palsy unspecified (76) | Intentional self-harm by strangulation and suffocation (90) |
| 2 | Epilepsy unspecified (10) | Unknown cause of mortality (59) | Epilepsy unspecified (60) | Asphyxiation (90) |
| 3 | Unknown cause of mortality (8) | Accidental poisoning by and exposure to narcotics and psychodysleptics (45) | Pneumonitis due to inhalation of food and vomit (39) | Unknown cause of mortality (62) |
| 4 | Neuronal ceroid lipofuscinosis (6) | Accidental poisoning by and exposure to antiepileptic, sedative, hypnotic, antiparkinsonian and psychotropic drugs (15) | Respiratory failure unspecified (28) | Accidental poisoning by and exposure to narcotics and psychodysleptics (37) |
| 5 | Other cerebral palsy (5) | Car driver injured in a collision with car, pick-up truck or van (11) | Pneumonia unspecified organism (27) | Unspecified injury to face and head (29) |
| 6* | Bacterial infection unspecified (<5) | Car driver injured in a collision with fixed or stationary object (11) | Sepsis unspecified organ (18) | Other psychoactive substance dependence (18) |
| 7* | Mucopolysaccharidosis type II (<5) | Epilepsy unspecified (11) | Bronchopneumonia unspecified organism (10) | Unspecified multiple injuries (18) |
| 8* | Other generalised epilepsy and epileptic syndromes not intractable with status epilepticus (<5) | Malignant neoplasm of the brain unspecified (10) | Acute lower respiratory tract infection unspecified (8) | Accidental poisoning by and exposure to antiepileptic, sedative, hypnotic, antiparkinsonian and psychotropic drugs (13) |
| 9* | Influenza due to other identified influenza virus with pneumonia (<5) | Pedestrian injured in a collision with car, pick-up truck or van (9) | Unknown cause of mortality (8) | Epilepsy unspecified (10) |
| 10* | Myotonic disorders (<5) | Assault by a sharp glass (8) | Other developmental disorders of scholastic skills (8) | Car driver injured in a collision with fixed or stationary object (10) |
| 11* | Pneumonitis due to inhalation of food and vomit (<5) | – | – | Car driver injured in a collision with car, pick-up truck or van (10) |
| 12* | Other deletions of part of a chromosome (<5) | – | – | Opioid dependence (10) |
| 13* | Sepsis unspecified organ (<5) | – | – | – |

*In the intellectual disabilities group (after the 5th top cause of underlying deaths), there were 8 causes of death from this point that had equal numbers per cause. For this reason, 13 causes have been included in this column. In the control group (after the 9th top cause of contributing causes of deaths), there were 3 causes of death from this point that had equal numbers per cause. For this reason, 12 causes have been included in this column.
ICD-10, International Statistical Classification of Diseases and Related Health Problems, 10th Revision.

age, sex, SIMD) for risk of death were as follows: avoidable, HR 5.2 (4.1–6.5) (adj HR 4.5 (3.6–5.7)); treatable, HR 16.1 (11.5–22.5) (adj HR 15.5 (11.0–21.8)); and preventable, HR 2.3 (1.6–3.2) (adj HR 1.9 (1.3–2.7)). The unadjusted rates were very similar to SMRs for avoidable, treatable and preventable risks of deaths although adjustment reduced the HRs slightly.

### Sex differences in the intellectual disabilities population only
When compared with controls, substantially higher differences were found in SMRs for females than males on all-cause, avoidable and preventable (but not treatable) mortality. SMR was calculated for the risk of mortality within the intellectual disabilities' population only, in which males were compared directly to females. No significant differences were observed between males and females in the intellectual disabilities population. The all-cause SMR was 1.2 (0.9–1.4); the avoidable SMR was 1.0 (0.7–1.4); the treatable SMR was 1.0 (0.7–1.5); and the preventable SMR was 0.9 (0.5–1.5). The median age of death in the intellectual disabilities population for males was 20 years (IQR=16–24) and 19 years (IQR=15–24) for females.

## DISCUSSION
### Summary/overview of principal findings
Our study makes an important contribution to understanding the relationship between intellectual disabilities and mortality. There have been very few studies on mortality in children and young people with intellectual disabilities, and most previous studies have been small in size and with inconsistent findings. Most did not provide granular, if any, information on cause of death nor avoidable deaths. We are aware of only one study that quantified avoidable deaths in children.[7] Our study brings crucial new insights into the extent of avoidable mortality, which previously has not been acknowledged. 33.8% of all the deaths in children and young people with intellectual disabilities were avoidable. Compared with the controls without intellectual disabilities, avoidable deaths occurred more commonly (131 vs 25 per 100 000 person-years), and both treatable (88 vs 5 per 100 000 person-years) and preventable (50 vs 22 per 100 000 person-years) deaths had substantially higher incidences.

We report that the median age at death for children and young people with intellectual disabilities was 19.5 years compared with 23.0 years for those without intellectual disabilities. 3.6% with intellectual disabilities died over the 9.5 years (399 per 100 000 person-years), compared with 0.3% without intellectual disabilities (36 per 100 000 person-years). SMR was 10.7; it was higher in females than males, higher in younger age groups and higher in more affluent neighbourhoods. For those with intellectual disabilities, the most common underlying and all-contributory causes of death were diseases of the nervous system, respiratory diseases and congenital/chromosomal abnormalities. In the control group, the most common underlying causes of death were external causes, symptoms/signs/abnormal clinical and laboratory findings, and neoplasms, and the most common all-contributory causes were external causes, injury/poisoning and diseases of the circulatory system. Where they could be calculated, SMRs were extremely high for all ICD-10 chapter groups of underlying (and all-contributory) causes, particularly so for diseases of the nervous system, respiratory system and digestive system.

### Comparison with existing literature and interpretations
Previous studies have reported higher rates of deaths in children and young people with intellectual disabilities, with SMRs ranging from 3.3 (95% CI 2.1 to 5.0) in young people aged 10–19 years[15] to 17.3 (95% CI 9.4 to 29.0) in young people aged 10–17 years.[8] Comparisons are limited, however, in view of the different age ranges studied. We found SMR to be higher at younger age, as has also been previously reported.[7 8 12] Variation in sample selection, definition of intellectual disabilities and sample size also limit comparisons. The closest studies in design to ours found an SMR of 21.6 (16.6–28.2) at ages 5–14 years, 7.7 (5.9–10.2) at ages 15–24 years[7]; and 30.4 (18.3–47.5) at ages 0–9 years, 17.3 (9.4–29.0) at ages 10–17 years and 3.7 (1.8–6.8) at ages 18–24 years.[8] These are similar to our SMRs of 27.4 (20.6–36.3) at ages 5–9 years, 15.8 (12.7–19.8) at ages 10–14 years, 8.6 (6.9–10.7) at ages 15–19 years and 6.6 (5.1–8.6) at ages 20–24 years, though CIs are wide at the youngest age group, and across all age groups in one of the studies.[8] Our study size enables us to provide more granular detail on the effect of age compared with previous studies.

There have been few previous reports on causes of death for children and young people with intellectual disabilities. Our findings on the most common immediate and all causes of death by ICD-10 chapter are similar to those reported in a previous study at ages 5–24 years.[7] Regarding specific all-contributing causes of death, our findings were similar, with cerebral palsy, epilepsy and respiratory conditions being the most common. A further study reported that the most common underlying causes of death in young people aged 1–25 years were respiratory infection (34%), aspiration-related causes (9.8%) and cardiac-related causes (14.7%). They found rates were especially high in the study for those aged <5 years for accidents (11.0%) and those aged >5 years for epilepsy (10.7%).[12] The study did not provide comparable all-contributing cause of death findings and included children at younger ages than in our study which may account for some of the differences. A recent systematic review reported that among children with intellectual disabilities deaths from pneumonia are 26 times higher, and respiratory-related deaths are 55 times higher, than in other children. These differences become less significant among adults with intellectual disability.[31]

Two previous studies have reported on avoidable mortality among children with intellectual disabilities,[7 8] only one of which provided numerical data.[7] It reported

that 19% had avoidable deaths, 29.7 (19.2–46) per 100 000 compared with 7.8 (7.0–8.8) per 100 000 in children and young people without intellectual disabilities (SMR=3.6; 2.3–5.5). They reported that the majority of avoidable deaths were treatable, including epilepsy, pneumonia and neoplasms. The treatable mortality rate among children with intellectual disabilities was 23.8 (14.6–38.8) per 100 000, compared with 2.0 (1.6–2.5) per 100 000 among controls (SMR=11.5; 7.0–18.8). We found considerably higher rates of avoidable and treatable deaths. We have also reported on preventable deaths which also occurred more commonly in the children and young people with intellectual disabilities. The previous study used school data as a marker for intellectual disabilities,[7] which may well be an overinclusive measure, and may account for these differences. As such, we contest that the issue of avoidable deaths, both treatable and preventable deaths, is a more serious issue in children than has previously been acknowledged. A further study reported only on young people with intellectual disabilities aged 18+, and the methods of cohort identification (via hospital inpatient records) may have failed to include some of the population with mild intellectual disabilities.[24] They found no difference in the risk of preventable deaths between young people with mild intellectual disabilities and the control group, but found that treatable deaths were more common (OR=7.75; 4.85–12.39), with 55% attributed to epilepsy. Their lack of difference in preventable deaths is in keeping with previous reports in the adult population with intellectual disabilities[32]—it appears there is a difference between children and adults in this regard.

We were able to identify children aged 5–9 years had the highest risk relative to controls for all-cause, avoidable and preventable mortality. It is possible that children who died in the youngest age group were those with the most severe intellectual disabilities, although we cannot be certain as this level of detail is not available. For treatable mortality, children at 10–14 years had the highest risk relative to peers, though CIs were wide and overlap for the 5–9 year-olds and the 10–14 year-olds; and these two younger age groups, relative to controls, had higher risk of treatable deaths than did the two older age groups. The high risk of treatable mortality in children and young people may be associated with the accessibility of high-quality healthcare and communication during health encounters. Previous research has shown that adults with intellectual disabilities receive significantly poorer management of long-term conditions in primary care according to best practice indicators from the Quality and Outcomes Framework,[33] experience more avoidable hospital admissions, considered potentially preventable with high-quality primary healthcare, and face a number of barriers in accessing health services, compounded by communication difficulties, and organisational and social support limitations.[34] However, little is known about the healthcare of children and young people with intellectual disabilities. Understanding and addressing healthcare inequalities in children and young people with

intellectual disabilities is crucial to reducing the risk of early mortality among this population.

All previous studies[1 7 8 12 18 19] (except two[14 17]) that have reported SMRs by sex for children and young people have found a higher SMR in females than in males. Similar to these studies, we found SMRs were higher for females for all-cause mortality, and we additionally found this to be the case for avoidable and preventable mortality; however, this was not the case for treatable mortality, where males had a higher SMR. Moreover, to further investigate apparent sex differences, SMRs for children within the intellectual disabilities population only were compared by sex, and no difference was found. This is the first study, to our knowledge, to report such detailed findings regarding avoidable mortality and sex differences and mortality in children and young people with intellectual disabilities.

We believe this is the first study with children and young people that has investigated mortality data in relationship to extent of neighbourhood deprivation. We found that there was a gradual increase in the SMRs in more affluent neighbourhoods. This is due to the difference in the general population across extent of neighbourhood deprivation, rather than any difference across neighbourhoods in the population with intellectual disabilities, that is, the general population experience higher rates of deaths the more deprived the area they reside in, whereas for the children with intellectual disabilities there is little difference across the extent of their neighbourhood deprivation.

### Strengths and limitations

The major strengths of this study are its large size, that it includes an entire country's population with intellectual disabilities, a comparison group, and that there was systematic enquiry on everyone as to whether or not they had intellectual disabilities. The census question on intellectual disabilities was subject to cognitive question testing prior to use, to ensure it accurately captured it and was acceptable to the population. Additionally, intellectual disabilities were distinguished from specific learning disabilities. The census had a 94% uptake,[25] and the record linkage was successful for >92%, hence limiting bias. Death registration is a statutory requirement in Scotland. We believe this study may be unique in including the whole country population of children and young people with intellectual disabilities, linked to death data. The results are likely to be generalisable to other populations in high-income countries. We excluded preschoolers to help reduce potential misclassification of children with undiagnosed intellectual disabilities.

Limitations include that death data came from death certifications by many clinicians, and deaths are infrequently verified by postmortem. The census data do not specify whether a record of intellectual disabilities was reported by a person with intellectual disabilities or their proxy (eg, a parent/carer, spouse, etc) or specific types of intellectual disabilities (eg, Down syndrome) or the

severity of intellectual disabilities. So, it is possible that there were some reporting issues that we are not aware of and we do not know how these could potentially impact on outcomes. As data were provided from NRS as complete cases, it was not possible to discern the number of incomplete cases (inclusive of intellectual disabilities), and possible differences from the complete cases. For children under the age of 16, we expect all reports to have come from proxy reports by parents, but we do not know the extent of proxy versus self-reports for the young people and we are unable to check this. The death certificate data indicate that there is likely to be heterogeneity in the different types of intellectual disabilities that may be included in our population, for example, cerebral palsy is frequently listed as an underlying cause of death. Intellectual disabilities may or may not co-occur with cerebral palsy. We expect that those who selected the option for intellectual disabilities in the census would include the population with intellectual disabilities and co-occurring cerebral palsy and exclude those with cerebral palsy with no intellectual disability. There were options for 'physical disability' and 'other condition' where the latter condition would be more appropriately placed. However, this level of detail about the reporting was not available. A related and important issue is the reporting of specific types of intellectual disabilities as an underlying cause of death. There are different perspectives on the accuracy of this practice,[35][36] with unique challenges that have not been but need to be discussed in research examining death certificate data for children. We have tried to balance these challenges by reporting the underlying causes of death as well as the contributing causes of deaths. This allows us to see the bigger picture in terms of those who were initially recorded with an underlying cause of death from a specific type of intellectual disability.

The rates and age-standardised SMRs using 5-year age bands for avoidable, treatable and preventable mortality were calculated using robust errors, except where there were fewer than 10 deaths per chapter. In keeping with the ONS methodology for investigating avoidable mortality,[7][8] all CMRs per 100 000 people based on fewer than 20 deaths were labelled as unreliable to warn users of the low reliability. It is also important to note that the ONS list of avoidable deaths is based on general population data and is possibly therefore an underestimate of avoidable deaths in child population with intellectual disabilities due to differing health and death profiles.

### Implications for practice, policy, education and research
Children with intellectual disabilities were more likely to die from all-cause, avoidable, treatable and preventable mortality than their peers, although the largest differences were found for treatable mortality, which may peak during late childhood. The high rates of mortality found in childhood for people with intellectual disabilities may be an important contribution to the substantial age of death differential, of 20 years lower on average, compared with the general population. As previously discussed, mortality studies have often not reported separate results for childhood and adulthood. The results of this study indicate improvements in the care and treatment of children and young people with intellectual disabilities are urgently required to reduce avoidable mortality outcomes and increase survival rates in the population with intellectual disabilities across the lifespan. This is particularly indicated for the top causes of avoidable mortality among children with intellectual disabilities in this research, including epilepsy, respiratory illnesses and digestive disorders. These conditions may present differently in people with intellectual disabilities and impact differently on mortality. It is vital that we better understand how each of these conditions influences people with intellectual disabilities specifically to identify the best pathways to initiate positive changes in healthcare and beyond. More research to understand why people with intellectual disabilities are dying disproportionately from avoidable deaths from these specific causes to inform future interventions is needed. Research attention should be directed to management of epilepsy, epilepsy risk assessments and multidisciplinary management on swallowing, feeding and posture to reduce aspiration/reflux/choking and respiratory infection that are leading to premature deaths in this population. Professionals working across different settings should be targeted for interprofessional collaboration to prevent pneumonia and complications from pneumonia in children and young people with intellectual disabilities, and to improve epilepsy care. Carers should be better informed of specific risks for avoidable health outcomes, particularly related to epilepsy, respiratory illnesses and digestive disorders, and the importance of early presentation for those in their care to seek medical assistance where required. Practical solutions must be identified to help reduce avoidable deaths, such as developing better guidance and protocols for health professionals (eg, primary care physicians, paediatricians, dentists, physiotherapists, speech and language therapists and dieticians) to better understand and treat the healthcare needs of children and young people with intellectual disabilities. This is important across all neighbourhoods, and a focus of professional activity in more deprived neighbourhoods is not justified for this population. The census question which asked about intellectual disabilities via methods of 'self-identification' is an effective way of identifying a vulnerable population with specific healthcare needs that could be used more widely in future research. We should pay close attention to a wider range of health/lifestyle-related factors that may increase or mitigate risks such as oral health, posture, feeding-related issues and lifestyle factors. To bring about real changes, the findings should be used to raise awareness among those who work directly with children and young people with intellectual disabilities and families (eg, health and social care professionals, education, community services, advocates, third sector organisations, formal and informal carers) to improve healthcare and adjustments to reduce these inequalities.

**Author affiliations**
[1]School of Health and Wellbeing, Mental Health and Wellbeing, University of Glasgow, Glasgow, UK
[2]School of Health in Social Science, University of Edinburgh, Edinburgh, UK
[3]School of Health and Wellbeing, Public Health, College of Medical, Veterinary and Life Sciences, University of Glasgow, Glasgow, UK
[4]Promoting a More Inclusive Society, Dundee, UK
[5]School of Health and Wellbeing, General Practice and Primary Care, University of Glasgow College of Medical, Veterinary and Life Sciences, Glasgow, UK

**Acknowledgements** The authors would like to acknowledge the support of the eDRIS Team (Public Health Scotland), in particular David Clark, for their involvement in obtaining approvals, provisioning and linking data, and for the use of the secure analytical platform within the National Safe Haven.

**Contributors** LAH-M analysed the data, interpreted the findings and wrote the first draft of the manuscript. ER analysed the data, interpreted the findings and contributed to the manuscript. MF and DM developed the record linkage, analysed the data, interpreted the findings and contributed to the manuscript. KD, LW, FS, FB, JM, JDS, BDJ, MT and JP interpreted the data and contributed to the manuscript. S-AC, CM, AH and DK conceived the study, analysed and interpreted the data, and contributed to the manuscript. All authors approved the final version of the manuscript. CM acts as guarantor.

**Funding** This work was supported by the UK Medical Research Council (grant number: MC_PC_17217), Baily Thomas Charitable Fund and the Scottish Government via the Scottish Learning Disabilities Observatory.

**Disclaimer** The funders had no role in the study design, collection, analyses or interpretation of data, writing the report nor the decision to submit the article for publication.

**Competing interests** None declared.

**Patient and public involvement** Patients and/or the public were involved in the design, or conduct, or reporting or dissemination plans of this research. Refer to the Methods section for further details.

**Patient consent for publication** Not applicable.

**Ethics approval** This study involves human participants and was approved by Scotland's Statistics Public Benefit and Privacy Panel (reference: 1819-0051) and the University of Glasgow College of Medical, Veterinary and Life Sciences Ethical Committee (reference: 200180081). Data sharing agreements are in place with the data controllers of all the linked data sets. Participants gave informed consent to participate in the study before taking part.

**Provenance and peer review** Not commissioned; externally peer reviewed.

**Data availability statement** No data are available.

**ORCID iDs**
Laura Anne Hughes-McCormack http://orcid.org/0000-0001-9498-8045
Sally-Ann Cooper http://orcid.org/0000-0001-6054-7700
Maria Truesdale http://orcid.org/0000-0003-3081-7858
Angela Henderson http://orcid.org/0000-0002-6146-3477

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
