## [Reviewer comments · BMJ Open]

ARTICLE DETAILS

TITLE (PROVISIONAL)	Rates, causes and predictors of all-cause and avoidable mortality in 163,686 children and young people with and without intellectual disabilities: A record linkage national cohort study
AUTHORS	Hughes-McCormack, Laura; Rydzewska, Ewelina; Cooper, Sally-Ann; Fleming, Michael; Mackay, Daniel; Dunn, Kirsty; Ward, Laura; Sosenko, Filip; Barlow, Fiona; Miller, Jenny; Symonds, Joseph; Jani, Bhautesh; Truesdale, Maria; Kinnear, Deborah; Pell, J. P.; Henderson, Angela; Melville, Craig

VERSION 1 - REVIEW

REVIEWER	Landes, Scott Syracuse University, Department of Sociology and Aging Studies Institute
REVIEW RETURNED	28-Feb-2022

GENERAL COMMENTS	Overall, I am very fond of this study and am glad the authors are addressing the topic. The use of the linked Scottish records data is well-suited for the study topic of mortality rates between children/young people with/without intellectual disability. My concerns are primarily regarding some methodological decisions in the study. 1. My primary concern is the measurement of the population of interest. As I read through the results, it was clear it included children/young people with intellectual disability, some of whom also had autism. This needs to be more clearly specified in the description of the measurement which I walked away from thinking it was just intellectual disability without autism. Also, based on the reporting of the causes of death, some of these individuals had cerebral palsy, and based on the description of the census data, some likely had Down syndrome. As there is evidence of heterogeneity in cause and age at death by type of intellectual and developmental disability, it will be important to be clear about who is/is not included in the measurement, and to sufficiently address the limitation of not being able to more fully differentiate by disability type in the limitations section. Finally, how were the cases with autism by no intellectual disability classified? This in no way lessens the contribution of this study, but needs to be more thoroughly defined and addressed throughout. 2. I would like to see sensitivity analysis comparing the complete case sample with excluded cases to ensure that this decision did not bias results. If it did not, please include these in the response with a description that they were checked in the text of the paper. If these cases were markedly different, I would include this analysis as an appendix.
--

	3. What was considered an error or ambiguity in cause of death records. One item that I would certainly consider an error would be the reporting of an intellectual or developmental disability as the underlying cause of death. Based on the description of the results stating cerebral palsy was listed as a cause of death, it appears the authors may not agree. In light of the literature describing this as an inaccurate practice in need of revision, I think it necessary to both clearly explain what was/was not included in the categories of error or ambiguity, and provide justification for why reporting cerebral palsy, or any other IDD, as the cause of death was seen as accurate. 4. The fact that you can likely estimate the percentage of individuals with intellectual disability and cerebral palsy among the deceased, but not the living should be discussed. It appears that cerebral palsy was a prominent factor (19%) in the intellectual disability deaths. I really see this as a distinct disability. This topic should be addressed on the front and back end of the paper. 5. I would find it more informative to see the ID category differentiated by whether the individual did/did not have a physical disability. While it looks like you would not be able to identify the actual type of physical disability, my guess would be that the mortality rates would be higher among this group as opposed to those without. I think this could be provided in an appendix as sensitivity analysis. 6. Regarding the end of the first para – our recent studies provide evidence that the age of death differential has reduced over time in the US.
--	--

REVIEWER	McCallion, Philip Temple University
REVIEW RETURNED	06-Mar-2022

GENERAL COMMENTS	Very important paper and makes great use fo existing datasets  1. Intellectual disability is reported by person or by proxy - paper does indicate that the census item was piloted but perhaps in the limitations there could be more acknowledgement that there may be under-reporting particularly for people with mild intellectual disabilities raising potential that causes may be skewed 2. More clarity on how people with autism (with and without intellectual disabilities) are included in the two samples 3. Paper notes general reports of twenty years less of longevity for people with intellectual disabilities - what do the rates of higher early deaths here mean for survival of adults with intellectual disabilities - might be addressed in the discussion 4. Again for the discussion - is it enough to say say we must pay more attention to pneumonia, respiratory and epilepsy or is there potential that these present differently for people with intellectual disabilities and the issue for future work is improper care rather than missed care?
---

VERSION 1 – AUTHOR RESPONSE

Reviewer: 1

Prof. Scott Landes, Syracuse University

Comments to the Author:

Overall, I am very fond of this study and am glad the authors are addressing the topic. The use of the linked Scottish records data is well-suited for the study topic of mortality rates between children/young people with/without intellectual disability. My concerns are primarily regarding some methodological decisions in the study.

1. My primary concern is the measurement of the population of interest. As I read through the results, it was clear it included children/young people with intellectual disability, some of whom also had autism. This needs to be more clearly specified in the description of the measurement which I walked away from thinking it was just intellectual disability without autism. Also, based on the reporting of the causes of death, some of these individuals had cerebral palsy, and based on the description of the census data, some likely had Down syndrome. As there is evidence of heterogeneity in cause and age at death by type of intellectual and developmental disability, it will be important to be clear about who is/is not included in the measurement, and to sufficiently address the limitation of not being able to more fully differentiate by disability type in the limitations section. Finally, how were the cases with autism by no intellectual disability classified? This in no way lessens the contribution of this study, but needs to be more thoroughly defined and addressed throughout.

Response: Thank you for these helpful questions and comments. We agree that the population needs to be outlined more clearly. The following has now been added to page 8:

“It is important to note that as multiple response options could be selected from the conditions list, the population with intellectual disabilities does overlap with the population with autism (as well as other conditions) and will include various different types of intellectual disabilities (for example individuals with Down syndrome or cerebral palsy [in cases where cerebral palsy co-occurs with intellectual disabilities]). This is however common in research of this nature as these two conditions often co-occur and there is also a range of causes of intellectual disabilities. The proportion of the population with intellectual disabilities who also reported co-occurring autism is reported in the results section.” (page 8)

In relation to the heterogeneity of the cause and age at death, the following has been added to the limitations:

“The death certificate data indicates that there is likely to be heterogeneity in the different types of intellectual disabilities that may be included in our population, for example, cerebral palsy is frequently listed as an underlying cause of death. Intellectual disabilities may or may not co-occur with cerebral palsy. We expect that those who selected the option for intellectual disabilities in the census would include the population with intellectual disabilities and co-occurring cerebral palsy and exclude those with cerebral palsy with no intellectual disability. There were options for ‘physical disability’ and ‘other condition’ where the latter condition would be a more appropriately placed. However, this level of detail about the reporting was not available.” (page 26/27)

In relation to the question regarding autism, we were able to remove any individual who reported autism only in the general population from the analysis.

“In the comparison population, individuals who reported autism (without also reporting co-occurring intellectual disabilities) were removed from the analysis as this population also experience a different health profile and substantial health inequalities compared to the general population, which would likely influence the mortality outcomes in the comparison population.” (page 8)

2. I would like to see sensitivity analysis comparing the complete case sample with excluded cases to ensure that this decision did not bias results. If it did not, please include these in the response with a

description that they were checked in the text of the paper. If these cases were markedly different, I would include this analysis as an appendix.

Response: The statement that was in the manuscript under missing data was misleading and has now been revised. The National Records of Scotland team who provided us with the data only provided us with complete cases. We do not unfortunately have more information about any incomplete cases that were removed before the data came to us. I have tried to clarify this better in the text as follows:

“Data linkage was conducted by NRS, and all data provided to us for this study included complete cases only. We included all the cases provided from NRS in the analysis. The only exception to this was if there was a date of death that came prior to the date of the census - these cases were excluded.”

3. What was considered an error or ambiguity in cause of death records. One item that I would certainly consider an error would be the reporting of an intellectual or developmental disability as the underlying cause of death. Based on the description of the results stating cerebral palsy was listed as a cause of death, it appears the authors may not agree. In light of the literature describing this as an inaccurate practice in need of revision, I think it necessary to both clearly explain what was/was not included in the categories of error or ambiguity, and provide justification for why reporting cerebral palsy, or any other IDD, as the cause of death was seen as accurate.

Response: This is an important issue and always presents a challenge in this type of research. There are different perspectives on this, and we have tried to balance this by reporting as standard the underlying causes of death (which do often include the type of intellectual disability e.g. Down syndrome or cerebral palsy) as well as the contributing causes of deaths. This allows us to see the bigger picture in terms of those who were initially recorded as an underlying cause of death from a specific type of intellectual disability. I have revised the section which mentions errors or ambiguity on death certificates to hopefully make it a bit clearer, as follows:

“Errors in cause of death records such as omission, use of abbreviations were listed as an unknown cause. All deaths where the underlying cause was ill-defined or defined by ICD-10 WHO guidelines³⁰ as codes in Chapter 18 were also re-classified as ‘unknown’.” (page 9)

I have also added to the limitations section to mention some of the issues around the points that have been made, as follows:

“The death certificate data indicates that there is likely to be heterogeneity in the different types of intellectual disabilities that may be included in our population, for example, cerebral palsy is frequently listed as an underlying cause of death. Intellectual disabilities may or may not co-occur with cerebral palsy. We expect that those who selected the option for intellectual disabilities in the census would include the population with intellectual disabilities and co-occurring cerebral palsy and exclude those with cerebral palsy with no intellectual disability. There were options for ‘physical condition’ and ‘other condition’ where the latter condition would be a more appropriately placed. However, this level of detail about the reporting was not available. A related and important issue is the reporting of specific types of intellectual disabilities as an underlying cause of death. This presents challenges in this type of research and there are different perspectives on the accuracy of this. We have tried to balance these challenges by reporting the underlying causes of death as well as the contributing causes of deaths. This allows us to see the bigger picture in terms of those who were initially recorded with an underlying cause of death from a specific type of intellectual disability.” (page 26/27)

4. The fact that you can likely estimate the percentage of individuals with intellectual disability and cerebral palsy among the deceased, but not the living should be discussed. It appears that cerebral palsy was a prominent factor (19%) in the intellectual disability deaths. I really see this as a distinct disability. This topic should be addressed on the front and back end of the paper.

Response: We hope we have now more clearly addressed this issue in the related sections below.

“It is also important to note that intellectual disabilities may or may not co-occur with cerebral palsy. It is likely that only individuals with cerebral palsy and intellectual disabilities would select the option for

'intellectual disabilities' in the census, and most likely in combination with 'physical disability'. Those with cerebral palsy with no intellectual disability may be more likely to select 'physical disability' (or possibly 'other condition'). However, this level of detail about the reporting was not available." (page 8)

And:

"The death certificate data indicates that there is likely to be heterogeneity in the different types of intellectual disabilities that may be included in our population, for example, cerebral palsy is frequently listed as an underlying cause of death. Intellectual disabilities may or may not co-occur with cerebral palsy. We expect that those who selected the option for intellectual disabilities in the census would include the population with intellectual disabilities and co-occurring cerebral palsy and exclude those with cerebral palsy with no intellectual disability. There were options for 'physical disability' and 'other condition' where the latter condition would be a more appropriately placed. However, this level of detail about the reporting was not available. A related and important issue is the reporting of specific types of intellectual disabilities as an underlying cause of death. This presents challenges in this type of research and there are different perspectives on the accuracy of this. We have tried to balance these challenges by reporting the underlying causes of death as well as the contributing causes of deaths. This allows us to see the bigger picture in terms of those who were initially recorded with an underlying cause of death from a specific type of intellectual disability." (page 26/27)

5. I would find it more informative to see the ID category differentiated by whether the individual did/did not have a physical disability. While it looks like you would not be able to identify the actual type of physical disability, my guess would be that the mortality rates would be higher among this group as opposed to those without. I think this could be provided in an appendix as sensitivity analysis.

Response: Thank you for this suggestion. We unfortunately did not have physical disability results in the specific data linkage set that we had access to (although this was collected in the census 2011). This would have been interesting and is something for us to consider adding for any future data linkages.

6. Regarding the end of the first para – our recent studies provide evidence that the age of death differential has reduced over time in the US.

Response: Thank you for drawing our attention to this. The first paragraph has been revised to include this information and reference, as follows:

"The life expectancy of people with intellectual disabilities has been reported to be shorter, on average 20 years younger than in the general population, although this may be substantially lower in some countries such as America⁴, including deaths considered potentially avoidable.⁵⁻⁶" (page 4)

Reviewer: 1

Competing interests of Reviewer: None.

Reviewer: 2

Prof. Philip McCallion, Temple University, University of Dublin Trinity College

Comments to the Author:

Very important paper and makes great use of existing datasets 1. Intellectual disability is reported by person or by proxy - paper does indicate that the census item was piloted but perhaps in the limitations there could be more acknowledgement that there may be under-reporting particularly for people with mild intellectual disabilities raising potential that causes may be skewed.

Response: Thank you for this comment. We should point out that in Scotland, children are assessed in educational services and receive additional resources as a result of recognition of having an additional support need (ASN). This is supported by the Additional Support Needs Act that was introduced in 2004 (and updated in 2009). This brings an advantage of extra support (targeted support in the classroom or enhanced support when extracted from the classroom) for children who are identified as having ASN. There is a legal requirement for schools to provide this support to children who have ASNs. So, in this sense, there is a benefit to seeking and confirming a diagnosis (for example of learning disabilities/ autism or specific learning difficulties such as Dyslexia/ Dyscalculia). Trends in educational data show there is likely overidentification of possible intellectual disabilities (and other conditions) among children in schools in Scotland, rather than under identification. This Act was brought in before the 2011 Census in Scotland from which this data is based on. We would therefore expect that under-reporting of children and young people with intellectual disabilities is unlikely, and if anything, there may be over-reporting. In the adult population, it would be more difficult to determine if there was under-reporting of mild intellectual disabilities, but we are confident that this is not an issue in the child population. We do appreciate that this is a complex issue, and an important point. We have added to the limitations to help elaborate on this issue, as follows:

“The Census data do not specify whether a record of intellectual disabilities was reported directly by the person with intellectual disabilities or their proxy (e.g., a parent/carers, spouse etc.) neither does it allow for the specification of types (e.g., Down syndrome) or severity of intellectual disabilities. So, it was not possible to report the differential impact on outcomes within different groups of people with intellectual disabilities.” (page 26)

2. More clarity on how people with autism (with and without intellectual disabilities) are included in the two samples.

Response: This is another important point that has also been raised by the previous reviewer. The following has been added to the manuscript for further clarification of this issue:

“It is important to note that as multiple response options could be selected from the conditions list, the population with intellectual disabilities does overlap with the population with autism (as well as other conditions) and will include various different types of intellectual disabilities (for example individuals with Down syndrome or cerebral palsy [in cases where cerebral palsy co-occurs with intellectual disabilities]). This is however common in research of this nature as these two conditions often co-occur and there is also a range of causes of intellectual disabilities. The proportion of the population with intellectual disabilities who also reported co-occurring autism is reported in the results section.” (page 8)

3. Paper notes general reports of twenty years less of longevity for people with intellectual disabilities - what do the rates of higher early deaths here mean for survival of adults with intellectual disabilities - might be addressed in the discussion

Response: Thank you for this suggestion. We have added a small section to the discussion in relation to this, as follows:

“The high rates of mortality found in childhood for people with intellectual disabilities may be an important contribution to the substantial age of death differential, of 20 years lower on average, compared with the general population. As previously discussed, mortality studies have often not reported separate results for children and adults. The results of this study indicate that improvements in the care and treatment of children and young people with intellectual disabilities are urgently required to reduce avoidable mortality and increase survival rates in the population with intellectual disabilities across the lifespan.” (page 27)

4. Again for the discussion - is it enough to say say we must pay more attention to pneumonia, respiratory and epilepsy or is there potential that these present differently for people with intellectual disabilities and the issue for future work is improper care rather than missed care?

Response: This is a really interesting point and we have added a section to the discussion to highlight this issue, as follows:

“This is particularly indicated for the top causes of avoidable mortality among children with intellectual disabilities in this research, including epilepsy, respiratory illnesses, and digestive disorders. These conditions may present differently in people with intellectual disabilities and impact differently on mortality. It is vital that we better understand how each of these conditions influence people with intellectual disabilities specifically to identify the best pathways to initiate positive changes in healthcare and beyond. More research to understand why people with intellectual disabilities are dying disproportionately from avoidable deaths from these specific causes to inform future interventions is needed.” (page 28)

Reviewer: 2

Competing interests of Reviewer: none

VERSION 2 – REVIEW

REVIEWER	Landes, Scott Syracuse University, Department of Sociology and Aging Studies Institute
REVIEW RETURNED	24-Jun-2022

GENERAL COMMENTS	My questions/concerns were thoroughly addressed. I have two remaining suggestions. I would amend this explanation in the limitations section “This presents challenges in this type of research and there are different perspectives on the accuracy of this. We have tried to balance these challenges by reporting the underlying causes of death as well as the contributing causes of deaths. This allows us to see the bigger picture in terms of those who were initially recorded with an underlying cause of death from a specific type of intellectual disability” to read “There are different perspectives on the accuracy of this practice, with unique challenges that have not been but need to be discussed in research examining death certificate data for children. We have tried to balance these challenges by reporting the underlying causes of death as well as the contributing causes of deaths. This allows us to see the bigger picture in terms of those who were initially recorded with an underlying cause of death from a specific type of intellectual disability. Regarding the first part of this statement – there are different perspectives – we had a healthy debate in DM&CN that can be referenced on this point if you see fit - https://onlinelibrary.wiley.com/doi/full/10.1111/dmcn.14221 and https://onlinelibrary.wiley.com/doi/full/10.1111/dmcn.14440. I would also add the fact that data provided as complete case to the limitations section as it was not possible to discern the number of incomplete cases inclusive of ID and possible differences from the complete cases. This is important especially with smaller size studies.
---

	I look forward to seeing this in print. It makes a substantial contribution to our understanding of the topic. Well done.
--	---

VERSION 2 – AUTHOR RESPONSE

Thank you for your consideration of the above named manuscript we submitted to the BMJ Open. We found the comments of reviewer 1 very helpful in improving our paper. We have addressed the points (received on 17/07/2022).